

# Achilles heel of a powerful invader: restrictions on distribution and disappearance of feral pigs from a protected area in Northern Pantanal, Western Brazil

Jose L.P. Cordeiro[1], Gabriel S. Hofmann[2], Carlos Fonseca[3] and Luiz Flamarion B. Oliveira[4]

[1] Fiocruz Mata Atlântica, Fundação Oswaldo Cruz (Fiocruz), Rio de Janeiro, RJ, Brazil
[2] Programa de Pós-Graduação em Ecologia, Universidade Federal do Rio Grande do Sul, Porto Alegre, RS, Brazil
[3] Departamento de Biologia & CESAM (Centro de Estudos do Ambiente e do Mar), Universidade de Aveiro, Aveiro, Portugal
[4] Departamento de Vertebrados, Museu Nacional, Universidade Federal do Rio de Janeiro, Rio de Janeiro, RJ, Brazil

Corresponding author
Jose L.P. Cordeiro,
jlpcordeiro@gmail.com

## ABSTRACT

This paper focuses on a rare case of natural disappearance of feral pigs (*Sus scrofa*) in an extensive area without using traditional methods of eradication programs. The study was conducted both in the Private Reserve of Natural Heritage (PRNH) Sesc Pantanal and in an adjacent traditional private cattle ranch. In 1998, feral pigs were abundant and widely distributed in the PRNH. However, the feral pigs gradually disappeared from the area and currently, the absence of pigs in the PRNH contrasts with the adjacent cattle ranch where the species is abundant. To understand the current distribution of the species in the region we partitioned the effects of variation of feral pigs' presence considering the habitat structure (local), landscape composition and the occurrence of potential predators. Additionally, we modeled the distributions of the species in Northern Pantanal, projecting into the past using the classes of vegetation cover before the PRNH implementation (year 1988). Our results show areas with more suitability for feral pigs in regions where the landscape is dominated by pastures and permeated by patches of Seasonal Dry Forest. The species tends to avoid predominantly forested areas. Additionally, we recorded that the environmental suitability decreases exponentially as the distance from water bodies increases. The disappearance of feral pigs in the PRNH area seems to be associated with changes in the landscape and vegetation structure after the removal of the cattle. In the Brazilian Pantanal, the feral pigs' occurrence seems strongly conditioned to environmental changes associated to livestock activity.

## INTRODUCTION

The different morphotypes of *Sus scrofa* Linnaeus—wild boar (javali), domestic (different breeds) and wild pigs (feral)—are the most widespread exotic ungulates in the world, with populations in demographic and spatial expansion in almost all Eurasian countries (*Fonseca & Correia, 2008*) and in most of the regions where they were introduced (Australia, South and North America). The species have achieved success in the conquest and occupation of foreign lands for centuries. Pig management and domestication probably began sometime between the 10th to 8th millennium BP in western Eurasia, and from then domesticated pigs were dispersed widely around the globe by humans (*Larson et al., 2007*). Currently, pigs are considered one of the world's worst invasive alien species (*Lowe et al., 2000*) and are present on all continents except Antarctica, and many oceanic islands (*Barrios-Garcia & Ballari, 2012*; *Long, 2003*). *Sus scrofa* have several biological traits and strong invasive abilities that allow them to occupy different habitat types throughout their exotic distribution range, thus making the eradication of this species (feral pigs) very difficult and expensive (*Mccann & Garcelon, 2008*; *Morrison et al., 2007*; *Parkes et al., 2010*). When compared to other ungulate species, wild boar show several attributes that are typical of r-strategists (*Geisser & Reyer, 2005*). They have the highest reproductive rate among ungulates, and their local density can double in one year (*Massei et al., 1997*). Additionally, the species has high ecological plasticity, a very opportunistic feeding behavior and a generalist approach to landscape use (*Gabor & Hellgren, 2000*; *Geisser & Reyer, 2005*).

In the Brazilian Pantanal (one of the largest continuous wetlands on the planet, covering approximately 140,000 km$^2$), *S. scrofa* introduction is believed to have occurred in the second half of the eighteenth century through traditional breeding of domestic pigs (*Alho & Lacher, 1991*). As reported in other areas of the world, pigs escaped from the ranches and became feral in a few generations through free reproduction in the wild (*Barrios-Garcia & Ballari, 2012*; *Bieber & Ruf, 2005*; *D'Eath & Turner, 2009*; *Dexter, 1998*; *Nogueira, Nogueira & Fragoso, 2009*). In 2000, the population of feral pigs was estimated at 10,000 individuals distributed throughout Pantanal (*Mourão et al., 2002*). In the Pantanal, the feral form is known as *porco-monteiro*. The species occurs primarily in open areas in seasonally flooded plains and near permanent lakes (*Alho et al., 2011*; *Desbiez & Keuroghlian, 2009*; *Desbiez et al., 2009*; *Keuroghlian, Eaton & Desbiez, 2009*; *Oliveira-Santos, 2013*). The species is strongly dependent on water bodies due to heat stress, which has been observed in other hot regions with periods of severe drought throughout the year (*Baber & Coblentz, 1986*; *Choquenot & Ruscoe, 2003*; *Dexter, 1998*; *Dexter, 1999*; *Mayer & Brisbin, 2009*). Although water is an environmental resource whose importance is obvious to most animal species, identifying important environmental parameters bounding species distributions is a complex task because animals respond to the environment at a range of spatial scales (*Turner et al., 1997*). Ungulates like feral pigs make foraging decisions both within and across a variety of spatial scales, making it difficult to relate species to specific habitats across their entire range (*Turner et al., 1997*). Therefore, the description of the species-habitat relationships is an important first-step towards understanding the linked ecological processes that can direct conservation decision-making, since the agents that determine population viability

may include factors related to habitat or elements that transcend spatial scales, such as dynamically linked variables or unlinked elements (*Hutchinson, 1978*; *Peterson et al., 2011*).

We present a rare case of natural disappearance of feral pigs in an extensive area without using traditional methods of control (eradication) programs. The drastic reduction in population of feral pigs occurred in a 14-year period (1998–2012) due to the transformation of cattle ranches into a Protected Area (PA). In PA there is no estimate of density or frequency of occurrence in previous periods, but the occurrence was common, as regularly observed by the reserve staff and by researchers (J Cordeiro & L Oliveira, pers. obs., 2014) during in a mammal survey in the region. In this context, feral pigs gradually "disappeared" from the PRNH. Park rangers report that visual records of feral pigs were extremely rare in recent past, with no records for years (RPPN Park rangers, pers. comm, 2014; J Cordeiro & L Oliveira, pers. obs., 2014). However, the current absence of pigs in PA area contrasts with the adjacent cattle ranch where the species is abundant. To understand the current distribution of feral hogs in the region we partitioned the effects of variation of feral pigs' presence considering the habitat structure (local), landscape composition and the occurrence of potential predators (jaguar and puma). Additionally, we modeled the distributions of feral pigs in Northern Pantanal, projecting into the past using the classes of vegetation cover before the PA implementation (year 1988). Our goal includes (i) identifying the spatial distribution patterns of feral pigs and (ii) inferring about the effect of landscape change, due to the implantation of a PA, in the occurrence of the species.

## METHODS

### Study area

The study was conducted in the municipality of Barão de Melgaço, state of Mato Grosso (MT), in the northeastern Brazilian Pantanal. The climate in the region is savanna type, "Aw," according to the Köppen's classification system (*Hasenack, Cordeiro & Hofmann, 2010*; *Hofmann, Oliveira & Hasenack, 2010*). Rainfall is concentrated in the austral summer and severe drought prevails in the rest of the year (*Nimer, 1979*). The region presents a flooding period from December through April, due to the accumulation of local rainfall and flooding of the headwaters of the Upper Paraguay River Basin (*Gonçalves, Mercante & Santos, 2011*). The herbaceous and woody vegetation in the region are influenced by the flooding regime adding variability to the landscape, characterized by a plain with low relief variability.

The data were collected in the Private Reserve of Natural Heritage (PRNH) SESC Pantanal, the largest private PA in Brazil (with 1,076 km$^2$) and in a traditional private cattle ranch (approximately 800 km$^2$). The two areas are adjacent and separated by the São Lourenço River (Fig. 1). The PA was established in 1998, after a long period of extensive livestock. Therefore, as other ranches in the region, PRNH contains exotic pastures cultivated in former areas of savanna or in forested areas cleared for pasture and artificial ponds for cattle (*Cordeiro, 2004*). However, after the removal of cattle in 1999, continuous monitoring of the landscape showed the gradual expansion of native forests in areas previously used by cattle (e.g., scrublands, pastures and earthmounds savannas)

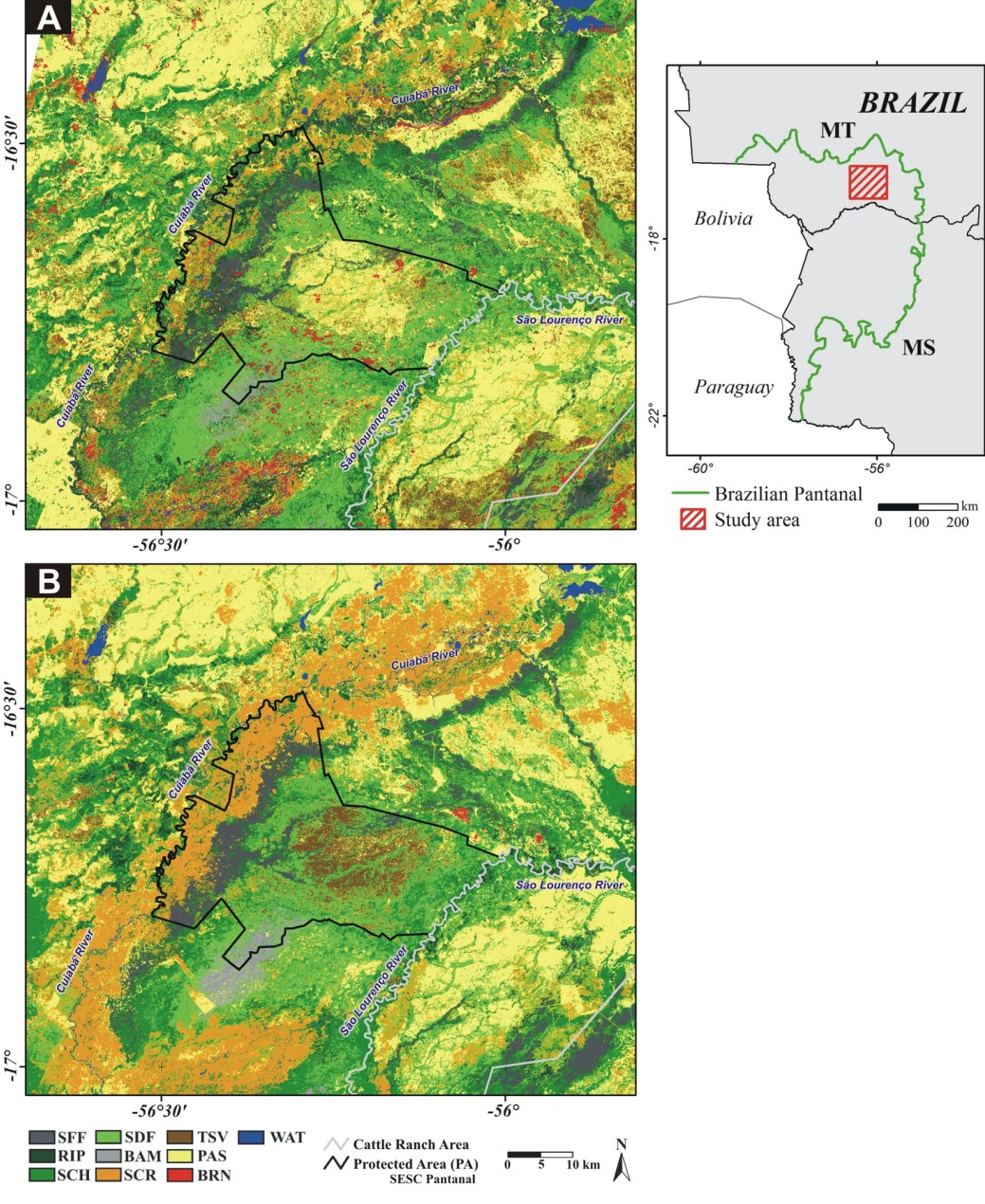

**Figure 1** **Study area location in Brazilian Pantanal,** *Mato Grosso* **(MT) and** *Mato Grosso do Sul* **(MS) states.** Land cover maps generated by Landsat image classification, (A) 1988 and (B) 2010. Seasonally Flood Forest (SFF); Riparian Forest (RIP); Scheelea Forest (SCH); Seasonally Dry Forest (SDF); Bamboo Forest (BAM); Scrubland (SCR); Termite Savanna (TSV); Pasture (PAS); Burned areas (BRN); Water (WAT).

(*Nunes da Cunha & Junk, 2004*; *Oliveira, Cordeiro & Hasenack, 2013*). Furthermore, with the removal of cattle and control of fire by PA staff, the open areas have undergone a succession process with a large increase in herbaceous/scrub vegetation density and biomass (*Oliveira, Cordeiro & Hasenack, 2013*).

## Occurrence data

Camera traps Reconyx PC90 High Output (Reconyx®, Inc., Holmen, WI, USA) were used to record feral pigs in the PA and in the cattle ranch (2010 and 2012). The cameras were programmed to operate in their standard module of motion sensitivity and pictures per trigger (three pictures, 1 s interval, no quiet period). No bait was used. The installation of camera trap sites was chosen randomly by direction and distance to be traveled (maximum of two kilometers) from roads (rivers, roads and trails) and considering the minimum distance of 600 m between sites. In the total, 180 sampling units were established (118 in the PA and 62 in the cattle ranch). The sampling effort per station varied from 15 to 28 days, totaling 3,862 trap days (2,559 in the reserve and 1,503 in the cattle ranch) and 92,688 h. Data were collected in both dry and rainy seasons. Additionally, we monitored 20 artificial ponds and 19 natural licks throughout 90 consecutive days, and we recorded all type of feral pigs' signs along the sampling campaigns. The camera trap survey in the PA, which resulted in a large record of the regional fauna, gives us support (*Hofmann et al., 2015*), confirming the absence or rarity of the species in the region.

## Feral pigs' relationships with landscape and habitat structure

We used two complementary methos, Variation Partitioning and Species Distribution Models (SDMs) was used to quantify the importance of environmental variables and to understand the reasons that led to the disappearance of this species in the PRNH. Variation Partitioning approach allowed us to compare the relative contribution of the variables (conditional or partial effect) and their independent effect (marginal effect) explained by factors in scales that are hardly addressed by methods such as SDMs (e.g., predation or structural components of vegetation measured at the local scale). On the other hand, SDMs approach allowed us to generate current (2010) and past (1988) potential distribution range using environmental factors associated with the areas currently occupied.

## Landscape characterization

We generated two land cover maps, for 1988 and 2010, based on satellite images classification (LANDSAT 5 TM, 27-Jul-1988 and 12-Oct-2010, with a spatial resolution of 30 m). The geoprocessing tasks were performed in Idrisi Taiga software (Clark Labs©, Worcester, MA, USA). Ten land cover classes were identified (Table 1; Figs. 1A and 1B). Two types of landscape descriptors were used: (i) the proportions of each land cover class; and (ii) the average distance to rivers or others water sources. The values were calculated by extracting the proportion of each class of cover or the average distances in the area formed by the buffers with a 500 m radius centered in each sampling unit.

## Variation partitioning analysis

We created an Index of Use (IU), for each species (*S. scrofa*, *Puma concolor*, puma, and *Panthera onca*, jaguar) at each site, considering the ratio between the time the species
**Table 1 Description of land cover classes of the study area.**

| Land Cover Class | Acronyms | Description |
|---|---|---|
| Scrubland | SCR | Open areas dominated by Byrsonima orbygniana, Hibiscus furcelllatus and Combretum lanceolatum |
| Seasonally Flood Forest | SFF | Monospecific forests dominated by *Vochysia divergens* "cambarazais" or by *Licania parvifolia* "corixos" |
| Termite Savanna | TSV | Fields with Curatella americana and rounded earthmounds covered by woody vegetation "murundus" |
| Seasonal Dry Forest | SDF | Forests with a predominance of deciduous trees such as Anadenanthera colubrina, Cedrela fissilis, Enterolobium contortisiliquum and Cordia glabrata |
| Scheelea Forest | SCH | Semideciduous forests where the understory is dominated by "Acuri" palm tree (*Scheelea phalerata*) |
| Bamboo Forest | BAM | Forest physiognomy with an emergent tree stratum and sparse understory dominated by "taboca" (*Guadua* sp.) |
| Riparian Forest | RIP | Unflooded forests that occur mainly on the banks of the São Lourenço River |
| Pasture | PAS | Herbaceous vegetation associated with intensive livestock , e.g., native and exotic grasses, and bare soil areas |
| Water | WAT | Water bodies such as rivers and lakes |
| Burned areas | BRN | Burned areas by the ranchers in order to clear land and increase the cattle stocking rates |

was recorded and the number of days the camera trap was active. After testing different time intervals (15, 30 min and 1 h) as a criterion for the independence of the records, we recognized that the longer periods resulted in an inflated index. We then considered consecutive shots of the same species at a maximum interval of 15 min as independent records. Likewise, at each sampling unit we evaluated the vegetation structure in five plots of 100 m$^2$, the first centered on the camera trap and the others at a 50 m distance in the four cardinal directions. We measured 11 attributes in each square (more details of the variables and methods used are provided in Table 2). The average values of the five plots were used to characterize habitat structure in the sampling unit.

We evaluated the size of the gradient through Detrended Canonical Correspondence Analysis (DCCA; *Ter Braak, 1986*; *Ter Braak & Smilauer, 2002*), considering the feral pigs' IU as a response variable in each sample unit and the predators IU, and habitat structure and landscape classes of cover (only data of 2010 land cover map) as environmental variables. Based on the length of the gradients we opted for Redundancy analysis (RDA), a linear method (*Ter Braak, 1986*; *Ter Braak & Smilauer, 2002*). We used the variation partitioning approach described by *Cushman & McGarigal (2002)*. Principal Component Analysis (PCA) was used to reduce the habitat structure and landscape data sets to seven and five uncorrelated components, respectively. The latent root criterion was considered to define the number of PCA axes to be used in the analysis (*Cushman & McGarigal, 2002*). We then submit the whole data set to forward selection (*Ter Braak & Smilauer, 2002*) to find a minimal set of variables that explain the species data about as well as the full set, and dropped all variables that were not significant at $p = 0.05$ to reduce collinearity among explanatory variables (*Cushman & McGarigal, 2002*; *Ter Braak, 1986*; *Ter Braak & Smilauer, 2002*). Noisy temporary predictive variables as burned areas were deleted from

**Table 2  Description of environmental variables and their units used as habitat structure metrics.**

| Variable (unit) | Description |
| --- | --- |
| Sky view factor (%) | Proportion of the sky hemisphere obscured by vegetation photographed at the center of each plot (*Frazer, Canham & Lertzman, 1999*). |
| Canopy height (m) | Canopy height estimated using a clinometers. |
| Dicots density (ind/ha$^{-2}$) | Number of individual flowering plants per unit area. |
| Basal area (m$^2$/ha$^{-2}$) | Area occupied by the cross-section of tree trunks and stems (CBH $\geq$ 5 cm) at breast height. |
| Dicots fruits (%) | Estimated by record of fruit in plots 5 (absence of fruit = 0%; registration in 1 plot = 20%, to record in 5 plots = 100%). |
| Palm fruits (%) | Estimated by record of palm fruit in plots 5 (absence of fruit = 0%; registration in 1 plot = 20%, to record in 5 plots = 100%). |
| Palm density (ind/ha$^{-2}$) | Number of individual palm trees per unit area. |
| Horizontal obstruction at ground level (%) | Average proportion of the profile board when viewed from across a distance of 5 m in the four cardinal directions (*Hays, Summers & Seitz, 1981*). |
| Horizontal obstruction at a height of 50 cm (%) | Average proportion of the profile board when viewed from across a distance of 5 m in the four cardinal directions (*Hays, Summers & Seitz, 1981*). |
| Horizontal obstruction at a height of 1 m (%) | Average proportion of the profile board when viewed from across a distance of 5 m in the four cardinal directions (*Hays, Summers & Seitz, 1981*). |
| Horizontal obstruction at a height of 1.5 m (%) | Average proportion of the profile board when viewed from across a distance of 5 m in the four cardinal directions (*Hays, Summers & Seitz, 1981*). |

the analysis. The PCA analyses were performed using Statistica 6.1 (StatSoft©, Palo Alto, CA, USA). RDA's analyses were performed in CANOCO for Windows 4.5 (*Ter Braak & Smilauer, 2002*).

## Species distribution models

We used maximum entropy niche modelling approach, as implemented in the MAXENT version 3.3.3k to describe environmental suitability, potential feral hog distributions and estimate the past distribution considering the environmental conditions. The method considers the requirement of the species based on the presence and on the set of environmental variables (*Phillips, Anderson & Schapire, 2006*), providing environmental variable response curves indicating how each variable affects the predicted distribution (*Phillips & Dudík, 2008*). We ran Maxent under the 'auto-features' mode and the default settings, with 10-fold replicates generated by bootstrap (*Phillips & Dudík, 2008*). The logistic output was used (habitat suitability on a scale of 0–1), with higher values in the Environmental Suitability Map (ESM) representing more favorable conditions for the presence of the species (*Elith et al., 2006*; *Phillips & Dudík, 2008*). For binary potential distribution maps (suitable/unsuitable), we applied the Minimum Training Presence (MTP) as a threshold value for model, because it is the most conservative threshold, identifying the maximum possible predicted area, while still maintaining a zero-omission rate for both training and test data.

The model was developed using 69 occurrence points (sampling units with *S. scrofa* presence) (Table S1) and ten (10) environmental variables (landscape descriptors). The program was configured to use 80% of occurrence data (56 points) for training and
20% (13 points) for test. The final models for feral pigs were based on the mean of the 10 replicated models. For the projection of the model to the past, to before the PA implementation, we used a 1988 land cover map (Fig. 1A). The Area Under Curve (AUC) of the Receiver Operating Characteristics (ROC) analysis was used as a measure of model performance (*Fielding & Bell, 1997*; *Manel, Williams & Ormerod, 2001*; *Peterson, Papes & Eaton, 2007*; *Phillips, Anderson & Schapire, 2006*). For comparative purposes, the resulting ESM images (2010 model and 1988 past projection), with continuous values from 0 to 1, were reclassified into five environmental suitability zones: (1) an Unsuitable Zone (USZ; value pixel suitability < Minimum Training Presence, MTP), (2) a Low Suitability Zone (LSZ, value pixel suitability between MTP value and 0.25), (3) an Intermediate Suitability Zone (ISZ, value pixel suitability between 0.25 and 0.50), (4) a High Suitability Zone (HSZ, value pixel suitability 0.50 and 0.75), and (5) a Very High Suitability Zone (VHSZ, value pixel suitability > 0.75).

Additionally, in order to quantify the spatial similarity between the model (2010) and its projection to the past (1988), we used Fuzzy index for continuous ESM, and Kappa index for binary maps (suitable/unsuitable). Both indices were implemented in Map Comparison Kit software, version 3.2.3 (*Visser & Nijs, 2006*) and express the pixel similarity for a value between 0 (fully distinct) and 1 (fully identical).

# RESULTS

## Variation partitioning

Abrupt changes in the landscape mosaic (DCCA; longest gradient shorter than 3.0) may affect the distribution of feral pigs. Landscape features and habitat structure explained 27.9% of the variation in feral pigs use in the study region (RDA model, $P = 0.001$) (Fig. 2). However, only one variable which describes habitat structure (first PCA axis related to sky view factor, basal tree area and canopy height) and three axes related to landscape features (Pasture, Scrubland and Seasonally Flood Forests cover; first, second and third axis, respectively) were included in the RDA model, based on the forward selection criterion ($P \leq 0.05$). Feral pigs do not seem to be conditioned on the use of the region by predators; this variable were not selected in the model. The first tier of the decomposition separated only habitat structure and landscape. The second tier decomposed feral pigs use variation, partitioning the landscape-level conditional effects by quantifying the unique explanatory power provided by each landscape variable: Pasture 19.7%, Scrubland 1.8% and Seasonally Flood Forests 1.7% (Fig. 2).

## Species distribution models

The model showed a very good overall performance, presenting high AUC values for both training (AUC = 0.932; SD = 0.010) and test data (AUC = 0.893; SD = 0.025), indicating that the modeled distribution performed better than the random one; high AUC denotes good observation/prediction fit of the test points in the spatial distribution model (*Lobo, Jiménez-Valverde & Real, 2008*). The most important environmental variable explaining the occurrence of feral pigs was Pasture (PAS), followed by scrubland (SCR) and Seasonal Dry Forest (SDF) (Fig. 3). The gain decreased most when the distance to water (WAT)

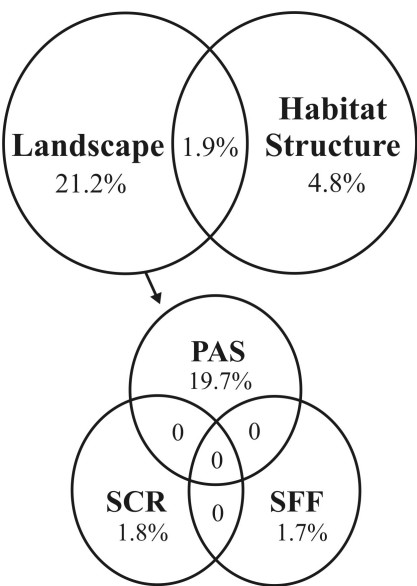

**Figure 2** **Venn diagram showing the first- and second-tier decompositions of feral hog RDA model ($P = 0.001$) in the study region.** The first-tier of the decomposition separates habitat structure and landscape contribution. The second-tier partitioned the landscape-level effect by quantifying the unique explanatory power provided by each landscape variable: Pasture (PAS), Scrubland (SCR) and Seasonally Flooded Forests (SFF).

was omitted, evidencing that this variable contains most of the information missing in the others (Fig. 3).

Feral pigs "prefer" (more suitability) landscapes dominated by Pastures (Fig. 4A), permeated by patches of Seasonal Dry Forest (Fig. 4C), with small portions of Scrubland areas (Fig. 4B), and areas with proximity to water sources (Fig. 4D). The species tends to avoid predominantly forested areas (SCH, RIP and SFF), and Termite Savannas (TSV).

The model indicates that the most suitable zones for feral pigs in 2010 are those located on large cattle ranches in the east, northeast and northwest of the protected area (Fig. 5A). Within the PA area, Low Suitability Zone (LSZ) predominate except in small isolated patches of intermediate and high suitability zones. On the other hand, the 1988 Environmental Suitability Map shows a #different scenario (Fig. 5C); the current PA area was filled with intermediate and high suitability zones contrasting with 2010. Figures 5B and 5D shows [represents] the potential distribution binary map (suitable/unsuitable) based on the MTP cutoff criteria (MTP = 0.09).

Between 1988 and 2010, there was a reduction of 84.1% in the suitable areas (suitability > MTP) within the PA area (Table 3), contrasting with the reduction of area occupied by these categories in the cattle ranch, which was less than 10%. The spatial and temporal similarities are shown in Table 3. Thereby, taking into account different criteria (Fuzzy for continuous values of suitability, and Kappa for binary maps- suitable/unsuitable), the PA area had the highest rate of change when compared to the cattle ranch and other areas of the study region.

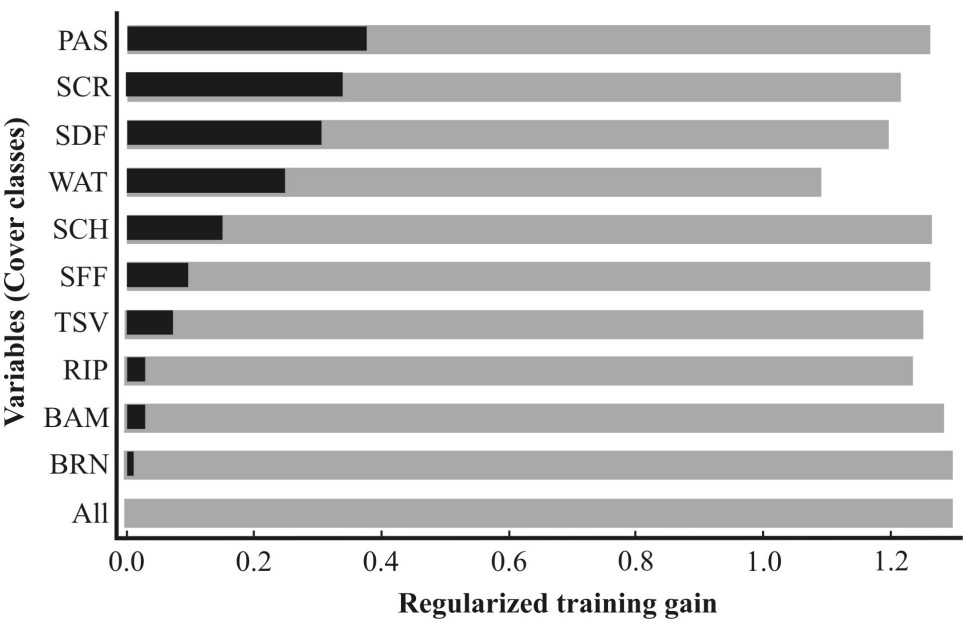

**Figure 3** Jackknife test results of individual environmental variable importance in the development of the MAXENT model relative to all environmental variables (dark grey bar) for each predictor variable alone (black bars), and the drop in training gain when the variable is removed from the full model (gray bars). Pasture (PAS); Scrubland (SCR); Seasonally Dry Forest (SDF); (WAT) Distance to Water; Scheelea Forest (SCH); Seasonally Flood Forest (SFF); Termite Savanna (TSV); Riparian Forest (RIP); Bamboo Forest (BAM); Burned areas (BRN).

## DISCUSSION

### Disappearance of feral pigs

The absence of feral pigs' records for many years in PA was intriguing, considering that we have been studying ungulates in the region since 1999 and we have many records of the species. *S. scrofa* is recognized for having large fluctuations of density and population size in native and exotic areas of occurrence (*Mayer & Brisbin, 2009*). Birth and mortality rates of young and adults are directly affected by the availability of food and environmental variations (*Baber & Coblentz, 1986*; *Bieber & Ruf, 2005*; *Geisser & Reyer, 2005*; *Jedrzejewska et al., 1997*; *Massei et al., 1997*; *Melis et al., 2006*; *Keuling et al., 2013*). However, due to their high reproductive potential, wild pigs are resilient, quickly recovering from such dramatic population reductions (*Mayer & Brisbin, 2009*). Sampling bias were discarded because both areas are part of a relatively similar ecological system in northern Pantanal, suggesting that the differences are not due to a drastic reduction in the ability to detect the species generating pseudo-absences (*Engler, Guisan & Rechsteiner, 2004*; *Morrison et al., 2007*). Nevertheless, despite a large additional sampling effort being put in areas potentially attractive for feral pigs we did not obtain any record of this species. In 2004, even with a much lower sampling effort feral pigs were recorded by camera traps in natural licks in this region (*Coelho, 2006*). Additionally, throughout the sampling campaign we traversed hundreds of kilometers across the area and we did not find any evidence or signs of feral
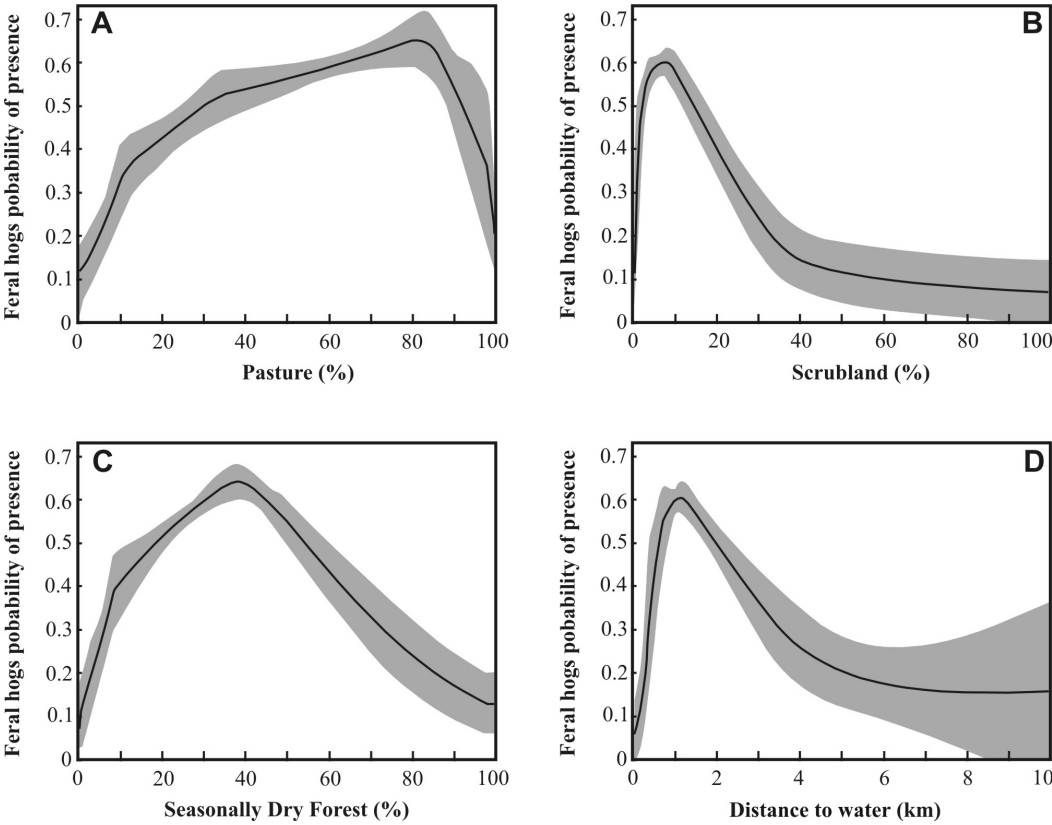

**Figure 4** **Response-curves of the variables in the *Sus scrofa* distribution model.** (A) Pasture (PAS); (B) Scrubland (SCR); (C) Seasonally Dry Forest (SDF); (D) Distance to Water (WAT). These curves show how each environmental variable affects the MAXENT prediction when all environmental variables are used to build the model.

pigs, such as wallowing sites, feces or tracks. We then assume that the lack of records in the PRNH Sesc Pantanal area is a real absence of feral pigs and not a pseudo-absence generated by the detection or sampling effort. Since we got only two records of feral pigs in the PRNH—both near the northeastern PA boundary, close to the limits of cattle ranch—and 261 records in the cattle ranch area between June and September 2012, we performed an intensive sampling campaign in areas potentially attractive for feral pigs in the PA monitoring 20 artificial ponds and 19 natural licks throughout 90 consecutive days. However, this sampling did not result in a single record of feral hog.

## Suitable habitats and limiting factors

Pastures were the most important land cover class related to the landscape feature, and they explain the distribution of feral pigs in the study region (RDA and SDM). This herbaceous class is maintained by grazing pressure (e.g., native grasses intensively grazed by cattle, exotic pastures, grasslands with very small earthmounds and bare soil areas). The intensive use of grasslands and pastures had already been described in southern Pantanal and in other regions (*Barrett, 1982*; *Baubet, Bonenfant & Brandt, 2004*; *Choquenot & Ruscoe, 2003*;

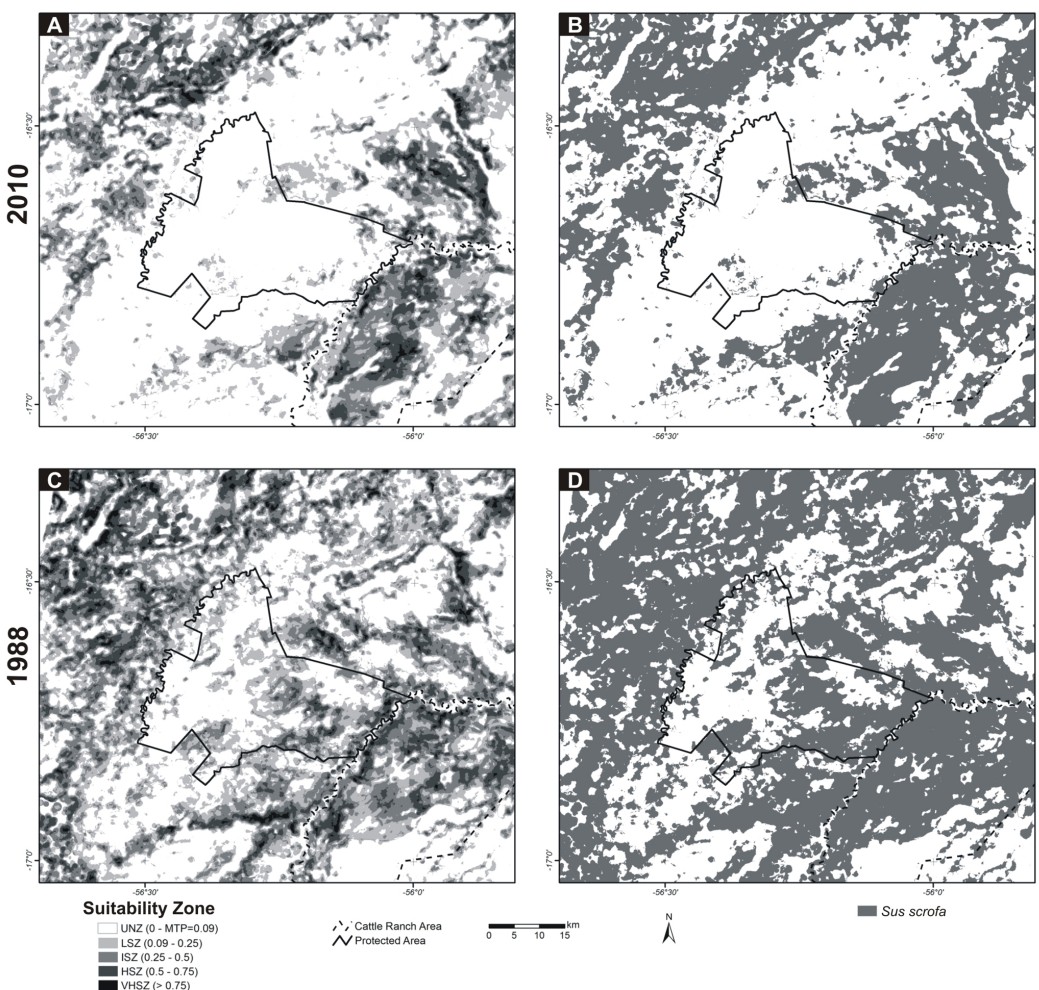

**Figure 5  MAXENT Environmental Suitability Maps for *Sus scrofa* in Northern Pantanal.** (A) 2010 model, (B) 2010 potential distribution binary map (suitable/unsuitable) based on the MTP cutoff criteria (MTP = 0.09), (C) model project to 1988 environmental conditions and (D) 1988 potential distribution binary map. Unsuitability Zone (UNZ), Low Suitability Zone (LSZ), Intermediate Suitability Zone (ISZ), High Suitability Zone (HSZ), and Very High Suitability Zone (VHSZ) identified.

**Table 3  The spatial and temporal similarities of suitable areas (> MTP[a]) for *Sus scrofa* in the study area.**

| Site | Area with value pixel suitability > MTP | | | | |
|---|---|---|---|---|---|
| | **1988** | **2010** | **Change rate** | **Fuzzy[b]** | **Kappa** |
| All study area | 539,122.4 ha | 332,871.8 ha | −38.3% | 0.33 | 0.24 |
| Protected area | 51,074.28 ha | 8,123.67 ha | −84.1% | 0.21 | 0.09 |
| Cattle ranch area | 78,714.5 ha | 73,677.2 ha | −6.4% | 0.41 | 0.26 |

**Notes.**
[a] Minimum training presence.
[b] For continuous map.

*Desbiez et al., 2009*; *Dexter, 1998*; *Dexter, 1999*; *Graves, 1984*; *Oliveira-Santos, 2013*), and plants like grass, herbs and forbs usually represent a considerable part of the feral pigs and wild boar diets (*Baber & Coblentz, 1986*; *Cuevas et al., 2013a*; *Cuevas, Ojeda & Jaksic, 2013b*; *Giménez-Anaya et al., 2008*; *Hellgren, 1993*; *Taylor & Hellgren, 1997*). Furthermore, the SDM approach showed that habitats with greater suitability for feral pigs are those predominantly herbaceous (around 80% coverage), interspersed with patches of seasonally dry forest (optimum between 35 and 40% coverage) and not too far from water bodies (around one kilometer). Feral pigs are known for their generalist habitat use (*Ilse & Hellgren, 1995*; *Mayer & Brisbin, 2009*) and by the preference for patchy habitats (*Acevedo et al., 2006*; *Gabor, Hellgren & Silvy, 2001*; *Oliveira-Santos, 2013*). The forest patches associated with a predominantly herbaceous matrix are very important because the dense vegetation is used as shelter from potentially lethal heat and as resting sites (*Choquenot & Ruscoe, 2003*; *Dexter, 1998*; *Graves, 1984*; *Huynh et al., 2005a*; *Huynh et al., 2005b*).

*S. scrofa* has a low tolerance to high temperature in nature due to the lack of sweat glands or other efficient physiological cooling mechanisms (*Baber & Coblentz, 1986*; *Choquenot & Ruscoe, 2003*; *Collin et al., 2001*; *Huynh et al., 2005b*), and low ability to concentrate urine (*Gabor, Hellgren & Silvy, 1997*; *Zervanos & Naveh, 1988*), being dependent on shaded habitats and reservoirs of water to avoid dehydration and promote thermoregulation (*Baber & Coblentz, 1986*; *Cuevas, Ojeda & Jaksic, 2013b*; *Dexter, 1998*; *Ilse & Hellgren, 1995*). Data available for southern Pantanal shows that the species has a high fidelity, returning to resting sites where they stay during the hottest hours of the day (*Oliveira-Santos, 2013*). We observed that away from forest patches feral pigs use small aggregations of trees (e.g., *Licania parvifolia*, *Couepia uiti* and *Calophyllum brasiliense*) and isolated small mounds with woody vegetation as thermal shelters.

The environmental suitability for feral pigs decreases exponentially as the distance from water bodies increases, as has already been observed for populations inhabiting arid and semiarid regions (*Baber & Coblentz, 1986*; *Cuevas, Ojeda & Jaksic, 2013b*; *Ilse & Hellgren, 1995*; *Mayer & Brisbin, 2009*). In arid regions of Australia, the species cannot persist in areas more than 10 km away from water sources, suggesting that the margin of their range is associated to inland river systems. Such areas vary temporally, acting as sources, pseudosinks and sinks (*Choquenot & Ruscoe, 2003*).

However, areas closer than 500 m to water bodies were not identified as highly suitable, particularly those close to rivers, reflecting the absence of feral pig records near the riverbanks within the study area. This may be due to the structure of the river banks, which have sharp slopes in long stretches. The low activity of feral pigs in riparian forests was already observed in southern Pantanal (*Keuroghlian, Eaton & Desbiez, 2009*). Habitats with close links with natural water bodies such as lakes and natural riparian forests were those with high records of jaguar in our study region. Jaguars have a close association with water in the Pantanal (*Crawshaw & Quigley, 1991*) and predation therefore could be an explanation for the lack of records of feral pigs in these habitats. These relationships could not be explored through the analysis, since the predator variable was excluded through the selection process. In any case, the existence of dozens of artificial ponds in the midst of extensive areas of pastures in cattle ranch region probably reduces the need of feral

pigs to access riparian forests where predation risk is higher, or even these areas can act as sinks. Likewise, the species showed negative relationships with structural features of the vegetation associated with forest habitats, such as tree density, and areas with closed canopy. The low biomass of grasses and herbs due to high shading caused by increasing canopy cover may be an additional explanation for the low utilization of riparian and the seasonally flood forests.

## Changes in the landscape features and loss of suitability areas for feral pigs

In Brazilian Pantanal, the occurrence of feral pigs seems to be closely associated with the environmental changes resulting from the land use by traditional livestock. The management system employed in the PA must have been the main factor that led to drastic reduction of suitable habitats and disappearance of feral pigs in this area. Important changes have been recorded over the last 40 years in plant communities and in the landscape of the Brazilian Pantanal by several authors (*Junk et al., 2006*; *Nunes da Cunha, Junk & Leitão Filho, 2007*; *Pott et al., 2011*; *Scremin-Dias, Lorenz-Lemke & Oliveira, 2011*). As from the seventies, the successions of wet years and large floods favored the colonization of tree and shrub species on the grasslands and pastures (*Nunes da Cunha & Junk, 2004*; *Nunes da Cunha, Junk & Leitão Filho, 2007*; *Pott et al., 2011*). Within this scenario, *Vochysia divergens* (Vochysiaceae) is the species whose expansion in the study region is most evident, although other species such as *L. parvifolia*, *Combretum lanceolatum*, *C. uiti*, *Byrsonima orbignyana* and *Ipomoea fistulosa* also have advanced over old open areas (*Nunes da Cunha & Junk, 2004*; *Oliveira, Cordeiro & Hasenack, 2013*; *Pott et al., 2011*). Since then, deforestation and 'controlled' fires have been the main forms of clearing land used by ranchers to increase cattle stocking rates (*Harris et al., 2005*; *Junk et al., 2006*; *Seidl, Silva & Moraes, 2001*; *Wilcox, 1992*). After the creation of the PA in 1999, thousands of cattle were removed from the area and a well-equipped fire brigade was established to control fires throughout the dry season (*Brandão et al., 2008*). With the absence large fires and grazing, the grasslands and other vegetation classes in the PA contrasted sharply with adjacent areas (*Oliveira, Cordeiro & Hasenack, 2013*). However, the environmental changes resulting from the traditional livestock are not restricted to the landscape scale, as they also affect the vegetation structure. Experiments of exclusion of cattle in the Pantanal showed that the absence of grazing pressure leads to strong growth of woody species (*Nunes da Cunha & Junk, 2004*). In the central area of the PA there was a huge increase in the biomass of grasses. The grasses reach about 1.5 m high and the access to many areas is difficult or almost impossible on foot or on horseback. In regions where grasses predominated a few years ago, in the floodplain of the Cuiabá River (western boundary of PA), shrubs dominate the landscape. In other words, nowadays the grasslands in the PA differ greatly from those occupied by feral pigs in the cattle ranch.

## Implications for conservation and management

Feral pigs are strongly associated with livestock. Over 80% (118,000 km$^2$) of Pantanal lands are cattle ranches and only 2.5% is formally protected in national and state parks

and in private protected areas (*Harris et al., 2005*; *Seidl, Davila & Silva, 1999*). Historically, many authors argue that traditional livestock plays an important role in the maintenance of the parkland physiognomy of the Pantanal and low density cattle ranching is considered an ecologically sound and sustainable management method (*Alho, Lacher & Goncalves, 1988*; *Junk et al., 2006*; *Pott & Pott, 2004*). Therefore, it is impossible to make an efficient plan for the conservation of the Pantanal without the inclusion of ranchers and their properties (*Harris et al., 2005*). Nevertheless, over the past 30 years the traditional livestock practices have been replaced by more intensive ones and ranchers have planted exotic pastures in forest areas cleared in order to increase cattle stocking rates (*Alho, Lacher & Goncalves, 1988*; *Oliveira, Cordeiro & Hasenack, 2013*; *Seidl, Silva & Moraes, 2001*). In the long run, however, these actions may be reversed against the ranchers. The reduction of natural areas and increase of environmental degradation due to the intensification of livestock in the region has certainly favored the growth of feral hog populations and this should result in large economic losses by damaged crops and husbandry (*Barrios-Garcia & Ballari, 2012*; *Bieber & Ruf, 2005*; *Gabor, Hellgren & Silvy, 2001*). Feral pigs are known in the Pantanal and in other regions to cause damage to large areas of grassland by foraging activity (*Desbiez et al., 2009*; *Mayer & Brisbin, 2009*; *Sicuro & Oliveira, 2002*). In the study area, large extensions of native and exotic pastures were completely wiped out (Fig. 6), with virtually no fodder for cattle or native grazers. Additionally, the high predation of eggs and native animals certainly are just some of the direct consequences of the increased density of feral pigs in these pasture areas (*Barrios-Garcia & Ballari, 2012*).

The disappearance of feral pigs in the PA area after the implementation of the management plan shows the vulnerability of the species and opens new possibilities for an eradication program in the region. The decline of feral pigs in this area appears to be intimately associated with the drastic reduction and fragmentation of pasture areas, a natural consequence of the fast succession of the vegetation after the cattle exclusion. Therefore, the loss and fragmentation of habitats by human actions, which are pointed to as major factors that lead to the extinction of species in global scale (*Banks-Leite, Ewers & Metzger, 2012*; *Cushman, 2006*; *Dobrovolski et al., 2013*; *Fahrig, 2002*) seem to have helped expel a powerful invader from the PA. The increased density and height of grasses due to the suspension of grazing cattle may also have a negative effect on feral pigs. Questions are open if the species disappearance is related to the reduction of habitat quality, low detection of predators, reduction of foraging efficiency or synergistic effect of various factors. In any case, changes in the land use regime, particularly in grasslands, can increase the chances of feral pigs management. Furthermore, a key factor to reduce feral pigs in areas with hot and dry season climate or semi-arid regions is to restrict their access to water sources (*Baber & Coblentz, 1986*; *Choquenot & Ruscoe, 2003*; *Dexter, 1998*; *Mayer & Brisbin, 2009*). Although it is virtually impossible to restrict the feral hog access to all water sources in a wetland like the Pantanal, increasing hunting in these places (especially those near to pastures and grasslands in dry seasons) as a major factor in limiting the size of the populations, targeting especially females and piglets, can be a way to keep the population in sub-optimal areas in order to facilitate management. The reduction in birth and survival rates by hunting focused on females and piglets can have a direct impact
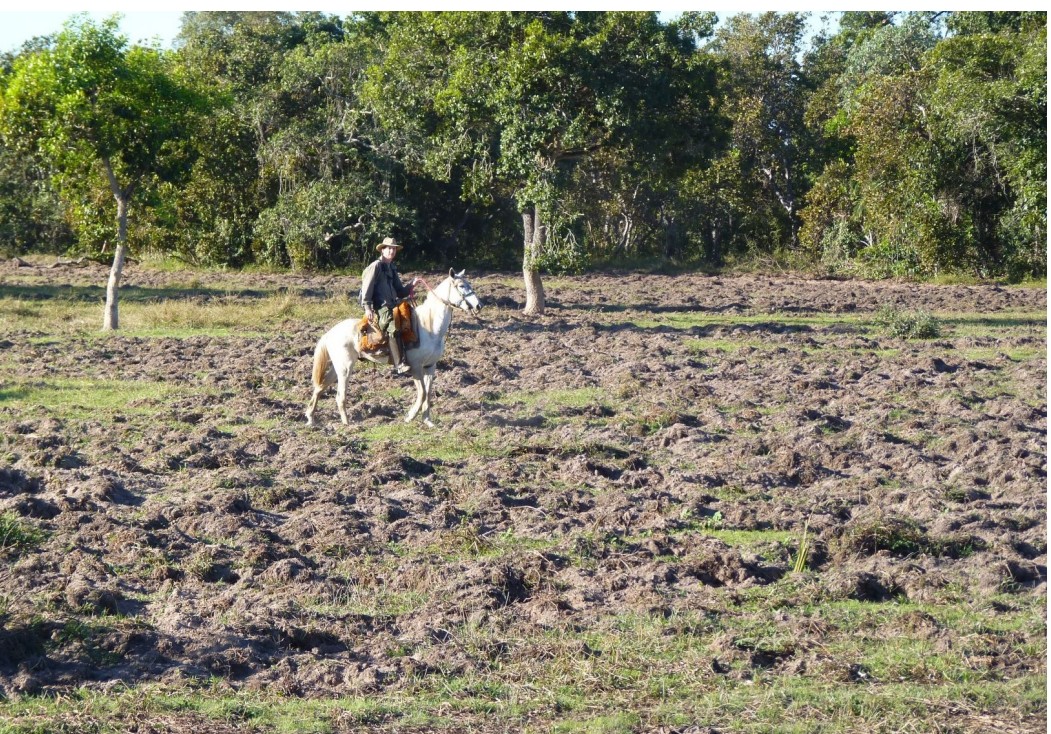

**Figure 6** **Ground rooting by feral pigs in northern Pantanal showing their activity in pasture areas.**
Photo credit: Luiz Flamarion Barbosa de Oliveira.

on local populations (*Bieber & Ruf, 2005*). *S. scrofa* is a highly cooperative and cognitive species. Under high hunting pressure survivors avoid techniques and sites targeted by hunters (*Morrison et al., 2007*) using their spatial memory to habitat selection, considering factors such as predation risk, thermal comfort and forage quality (*Oliveira-Santos, 2013*). Permanent hunting pressure near artificial ponds in pasture areas forces feral pigs to seek alternative sources of water, increases energy expenditure and reduces time spent in thermoregulation, hence forcing the use of less suitable habitats such as riparian forests increasing the risk of predation. Synergic interaction between several factors may be more important in limiting the population growth of the species in remote areas like Pantanal than simply directing efforts to a single method of population control.

## CONCLUSION

The disappearance of feral pigs after the conversion of old cattle ranches into a protected area was associated with changes in the landscape and vegetation structure after the removal of the cattle and the implementation of the management plan for the area. In the Brazilian Pantanal feral pigs occurrence seems conditioned to environmental changes associated to livestock activity, particularly related to the proportion of pastures available, although the availability of forest patches and water sources are also important. They are rare in continuous and riparian forests. Occurrences of potential predators are not significant. However, predation cannot be completely ruled out as an important factor in conditioning

the distribution of this species in Pantanal, but more data should be raised about this factor, in this region with varied mosaic and still rich fauna. Under current conditions the chances of recolonization of the protected area are low, particularly by the absence of suitable habitats. For the same reason it is hard to believe in a "resurgence" of feral pigs via "Lazarus effect" as refered as "Lazarus pig" (*Morrison et al., 2007*), the last animal in the region. Thus, our results suggest a point of weakness in the exotic distribution of *S. scrofa*, directly related to the ressurgence of grazed areas. The distribution of the species in the Brazilian Pantanal is the result of the effect of human activity on the structure and spatial arrangements of plant formations in the region.

## ACKNOWLEDGEMENTS

The authors would like to thank Sesc Pantanal (for providing logistic support and funding), and to all employees for their help with the collection of samples. We are especially grateful to Sesc managers Leopoldo G. Brandão, Maron E. Abi-Abib, Waldir Valutki, Cristina Cuiabália, and Silvia Kataoka. We thank Gustavo Staut for support and permission to work on the Sta. Lucia Ranch. We thank Igor P. Coelho, Vinicius A.G. Bastazini, and Fernando Sicuro, among other colleagues, for the support given during the field work.

### Funding

All the funding that supported this research was provided by two official Brazilian support agencies. The authors received a research grant from the Brazilian Research Council (CNPq - Grant 400713/2013-6), and Coordination for the Improvement of Higher Education Personnel (CAPES; fellowships to José Luis Passos Cordeiro, Proc.: BEX 1302/15-9; Gabriel Selbach Hofmann). Field work was partially supported by Sesc Pantanal. There was no additional external funding received for this study. The funders had no role in study design, data collection and analysis, decision to publish, or preparation of the manuscript.

### Grant Disclosures

The following grant information was disclosed by the authors:
Brazilian Research Council: CNPq - Grant 400713/2013-6.
Coordination for the Improvement of Higher Education Personnel.
Sesc Pantanal.

### Competing Interests

The authors declare there are no competing interests.

### Author Contributions

- Jose L.P. Cordeiro and Gabriel S. Hofmann conceived and designed the experiments, performed the experiments, analyzed the data, contributed reagents/materials/analysis tools, wrote the paper, prepared figures and/or tables, reviewed drafts of the paper.
- Carlos Fonseca wrote the paper, reviewed drafts of the paper.

- Luiz Flamarion B. Oliveira conceived and designed the experiments, performed the experiments, analyzed the data, contributed reagents/materials/analysis tools, wrote the paper, reviewed drafts of the paper.

## Data Availability

The raw data has been supplied in Table S1.

## Supplemental Information

Supplemental information for this article can be found online at http://dx.doi.org/10.7717/peerj.4200#supplemental-information.

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
