# Peer review of "Achilles heel of a powerful invader: restrictions on distribution and disappearance of feral pigs from a protected area in Northern Pantanal, Western Brazil"

_PeerJ, doi:10.7717/peerj.4200_

## Round 0.1 · original submission · Minor Revisions

The study deals with feral pig invasion, as one of the Revewer's mentioned this is a relevant problem worldwide and an important issue in Brazil . I agree with the reviewers that it is an interesting paper with a number of issues that should be clarified and corrected before publication. When revising your MS, I sincerely ask you to adequately address EACH of the referees' comments, by either incorporating the suggestions in the revision, if possible, or providing brief but convincing rebuttals in case you do not agree with them. Thank you for giving us the opportunity to consider your work.

Reviewer 1 ·

Basic reporting

The study deals with an important issue in Brazil (feral pig invasion) and with an interesting ecological question (why did the specie disappear in a Protected Area?) with considered applicability to conservation strategies (invasive species control). However, the study failed to properly describe the situation (Specie disappearance), there were misleading use of concepts (Potential niche x occurrence of specie), the methods lack more details for evaluation (RDA and SDM), and the analysis and results could be substantially improved:

1 –Specie disappearance:
The manuscript is ambiguous and unclear for the situation about the disappearance of the species in the PA. It is plausible the concentration of S.scrofa population has moved from PA to the adjacent cattle ranch due to the land use change. This phenomenon is really interesting, novel and useful for decision making regards to specie’s control. However, the specie’s condition was not clear to the readers till the Discussion section. Is strongly recommend to be fair with the readers since the beginning.
Actually, the specie didn´t disappear from PA as pointed out along the text (Title, Abstract and Introduction). At "Study area" section, it was clear the species still there: "Park rangers report that visual records of feral pigs were extremely rare and the species is seldom recorded". The Results section unfortunately failed to inform information about the presence/absence. By the other hand, the Discussion section started with "The low number of records of feral pigs in the PA..." and gave the impression the specie’s abundance in PA was low but it has not disappeared. The specie´s disappearance is a key situation for the study and it must be properly describe for the readers and for better evaluation.
Moreover, the apparent specie´s abundance reduction was based on the informal observation in the past in the two areas (PA and the adjacent cattle ranch). However, there were no measure of abundance for spatial and temporal comparison between them and the population concentration could be superficial, especially by considering the distance between the areas. The two areas are very close for the biology of the specie and it is likely there is only one population with the same variation in both areas. The population reduction in PA could happen in both areas and the study was unable to detect and demonstrate it. In regards of high sampling effort with few specie’s records, it is plausible to considered the current low S.scrofa abundance in PA, but there is no such information to the other area for proper comparison. The so called disappearance could be an overall phenomenon within the region, the municipality of Barão de Melgaço, without relationship with the PA effect.

2 - Potential niche x occurrence of specie
The effect of PA in occurrence of specie is the key issue for the study goals: “(i) identifying the spatial distribution patterns of feral pigs and (ii) inferring about the effect of landscape change, due to the implantation of a Protected Area (PA), in the occurrence of the species”. To these goals, the occurrence must be somehow measured in the field like occupancy (e.g., Mackenzie et al., 2006 - “Occupancy Estimation and Modeling” ). However, the study used the potential niche for this goal instead. The occurrence was then an extrapolation of the niche theory and not the proper measuring in nature. The study considered potential niche as “probability of presence” (Figure 4), i.e., a kind of “potential presence”. Therefore, the study was reduce to a merely description of the potential niche variation in the study area and it had limitations to the inference about the true PA effect on the specie occurrence.
Furthermore, the potential niche was dependent on 69 current records of the specie and it can results in biased estimation to the present and past niche/occurrence. The method has the assumption of unreal perfect detectability and it can easily be violated by false absence (e.g., Mackenzie et al., 2006 - “Occupancy Estimation and Modeling” ).
The data base for the potential niche modeling also need more details for better evaluation of this method as described below.
For this case and goals, I suggest using proper methods for occurrence other than potential niche, like occupancy models.

3 – More detail need for RDA and SDM
Multivariate analysis like RDA are proper evaluated with the complete results of axis and not only percentage of explained variation by each axis (Fig. 2). The complete results can be provided as table in supplementary material but it is also interesting to plot the results against the explanatory variable and ordering them to highlight the main effect.
Species Distribution Models (SDM) used in this study was based in 69 records of the species without more details. It fundamental to provide more details for better of evaluation of this method as such its geography distribution (e.g., in the map, Fig. 1) and how they were collected (camtraps only or all data of field effort like the monitored natural licks and artificial pounds). I suggest providing this information on the text and, if it was the case, I discourage the use of data based on the monitored natural licks and artificial pounds because they can work as attractive site and biased the presence and then the potential niche estimation.

4 – Analysis and results improvements
As many other invasion biology case, the human dimension is an important issue. However, the study was limited by considering only ecological variables. The results can be very biased because the variable comparison is based on relative importance. I suggest including in the analysis a human variable like the distance of settlement or other one that represent the influence of human mediated invasion.
For PA effect, the cattle was removed as well as the human presence with their culture (wild pig management) and livestock (free range system of domesticated pigs). The feral pigs in Pantanal is reported and known as an abundant and old population, but it still unclear how much the real human presence conserve this population by local management with constant introduction (free range of domesticated pigs) and cultural hunting (Desbiez et al., 2011). This alternative interpretation does not reject the Authors´ one, but the article could have interesting improvement by considering human influence in this invasion biology. So far, the analysis have the assumption this invasion is strong dependent of biology of the species (dependent of "changes in the landscape and vegetation structure"), but the indirect human influence can much stronger. The management of cattle changes the landscape, but it also means constant human presence with their externalities as such conserving the wild pig population in higher abundance with protection (captive pigs close to the house) followed by constant restocking (scapeing pigs) and biased harvest (preferable male hunting and some female care). The Achilles heel of S.scrofa can be the amount of propagule and restocking. An extra variable considering the human influence can improve the analysis.

Experimental design

The Index of Use (IU) has several limitations for the analysis and it could be replaced by other index or analysis. The first limitation is the spatial dependence of the camtrap sites. The minimum of 600 m is not enough for S.scrofa, puma and jaguar because the species has wider home range in 15-28 days. The same animal can use several sites and overestimate the IU. An alternative is to test the dependence for each species reducing the intervals as it was done for shots intervals. I suggest at least justifying clear the reason to use the sampling interval in each site.
The other limitation of IU was the assumption of the perfect and constant detection within and among species. The Authors should consider this index (IU) has the naïve assumption of perfect detection, i.e. when the species are presence at the site, it is always recorded and it scores the index; otherwise when the specie are absent the specie is not recorded. Thus, there is no false absence. The detection is also equal for all sites and all species has the same detection probability. However, it is expected different detection probability among sites, i.e. difference between forest and open area, for example. In the same manner, jaguar sounds more difficult to detect than puma and S.scrofa. It is very important then to control the detectability for this analysis. For this issue, I suggest the use of occupancy models (Mackenzie et al., 2006 - “Occupancy Estimation and Modeling” ).

Validity of the findings

The findings were very interesting but dependent on the kind of variables (only ecological variable) and spatial dimension for the species.
It is important to consider human variable to balance the relative importance among ecological variable. Such variables can be justified by the conflict of interest and it has important implications for controlling the specie in other parts of Pantanal, as the Author justify in the Introduction section ("direct conservation decision-making"). The conflict of interest was an important reason to consider the S. scrofa among the 100 of the world’s worst invasive alien species (Lowe et al., 2004) and its control depend on the overcome of this feature (Choqnot et al., 1996, Lowe et al., 2004). Without controlling the human influence, it is difficult to eradicate feral population.
The findings (“strong relationship with pasture”) can also be biased by the spatial dimension of the study because it was very narrow for the species. The results can just be a local effect of abundance variation or reduction of the abundance in the periphery of the local population´s range. In wider geography scale, the findings was not expected because the invasion is very old (>200 year old) and the species had been restricted to the Pantanal biome for centuries even with land use change around to open area like pasture-like system. For some reason, the species have not invaded the rest of Brazil even with enough time and favorable land use change.
These considerations don’t compete with the original propose. They are just a complement. The suggestion can change the analysis but improve the interpretation, match the real invasion situation of the species in this country and provide more insights for decision making.

Reviewer 2 ·

Basic reporting

The study deals with feral pig invasion, a relevant problem worldwide and an important issue in Brazil . The authors, properly describe the past and current situation at the study area, and try to answer very interesting ecological questions applying proper research methods and analysis. Adittionaly, the study is very important for conservation strategies and invasive species control.

Experimental design

Like in most biological invasions, in the studied case, the human dimension is an important topic. However, the study was focused only in ecological variables. The importance of invasion history is also underestimated. It would be relevant to include in the analysis historical data about the micro invation within Pantanal, and a human variable like the distance of settlement or other one that represent the influence of human assisted invasions in the modelling.
Also, I recommend that, whenever feasible, similar macro-scale approaches to the species distribution modeling are pursued to capture the extent of conditions that support the species populations and generate hypotheses about species limitations or invasion potential that can be tested in combination with this type of finer-scale research.

Validity of the findings

Although long-term movement data for wild pigs is lacking in South America, social factors such as value of wild pigs as a recreational hunting resource or as farmed species are at least as important as natural dispersal in driving the current distribution of wild pigs. As such, we recommend that future research investigating the distribution and invasiveness of wild pigs should include social factors in addition to biological factors (what biotic and abiotic factors limit wild pigs populations?) to address competing hypotheses and generate effective management solutions.
Despite of this, I strongly recommend the publication of this very interesting manuscript.

---

## Round 0.2 · accepted · Accept

The authors have made all changes suggested by the reviewers and the editor. Some minor editing changes are included in the attached version.